# Crystal structures of the human neurokinin 1 receptor in complex with clinically used antagonists

Jendrik Schöppe [1], Janosch Ehrenmann [1], Christoph Klenk [1], Prakash Rucktooa[2], Marco Schütz[1,3], Andrew S. Doré[2] & Andreas Plückthun [1]

Neurokinins (or tachykinins) are peptides that modulate a wide variety of human physiology through the neurokinin G protein-coupled receptor family, implicated in a diverse array of pathological processes. Here we report high-resolution crystal structures of the human $NK_1$ receptor ($NK_1R$) bound to two small-molecule antagonist therapeutics – aprepitant and netupitant and the progenitor antagonist CP-99,994. The structures reveal the detailed interactions between clinically approved antagonists and $NK_1R$, which induce a distinct receptor conformation resulting in an interhelical hydrogen-bond network that cross-links the extracellular ends of helices V and VI. Furthermore, the high-resolution details of $NK_1R$ bound to netupitant establish a structural rationale for the lack of basal activity in $NK_1R$. Taken together, these co-structures provide a comprehensive structural basis of $NK_1R$ antagonism and will facilitate the design of new therapeutics targeting the neurokinin receptor family.

---

[1] Department of Biochemistry, University of Zürich, Winterthurerstrasse 190, CH-8057 Zürich, Switzerland. [2] Sosei Heptares, Steinmetz Building, Granta Park, Great Abington, Cambridge CB21 6DG, UK. [3] Present address: Heptares Therapeutics Zürich AG, Grabenstrasse 11a, 8952 Zürich, Switzerland. These authors contributed equally: Jendrik Schöppe, Janosch Ehrenmann. Correspondence and requests for materials should be addressed to A.P. (email: plueckthun@bioc.uzh.ch)

The neurokinin (NK) receptor-ligand system represents a complex, evolutionarily conserved neuropeptide signaling architecture[1,2]. Derived from alternate processing of two genes, the best-characterised mammalian neurokinins are Substance P (SP), NKA and NKB, sharing the conserved C-terminal structural motif FxGLM-NH$_2$[3]. Among other neurokinins, these three peptides act as agonists with different affinities and selectivity for three pharmacologically distinct neurokinin receptors[4] (NK$_1$R, NK$_2$R and NK$_3$R) that belong to the superfamily of G protein-coupled receptors (GPCRs). Within this system, SP represents the preferred endogenous agonist of NK$_1$R[4]. NK$_1$R has been shown to be present in the central and peripheral nervous system[5,6], smooth muscle[7], endothelial cells[8] and also on cells that participate in the immune response[9]. Over the past four decades, intensive research has linked the SP-NK$_1$R system to such diverse pathophysiological processes as nausea[10], analgesia[11,12], inflammation[13], pruritus[14] and depression[15,16], highlighting the potential therapeutic value of antagonists directed against NK$_1$R. This prospect triggered widespread efforts across industry and academia to discover such compounds to date[17,18].

Disclosure of the first non-peptide NK$_1$R antagonist CP-96,345[19] (Supplementary Figure 1), discovered by high-throughput screening (HTS), subsequently spurred the development of a number of antagonists with improved pharmacological properties. This led to the identification of CP-99,994[20], which reduced the chemical structure of CP-96,345 to a molecular scaffold found in many later-stage small-molecule antagonists. CP-99,994 displays high affinity and selectivity for NK$_1$R, as well as efficacy in animal models, and therefore historically provided a valuable pharmacological tool for the investigation of the physiological role of SP-mediated signaling through NK$_1$R[21]. Modification of the central saturated six-membered piperidine ring of CP-99,994 alongside further scaffold optimisations[22,23] (by substituent addition and modifications) ultimately lead to the development of aprepitant[24] (Supplementary Figure 1), which became the first approved oral drug to make it into the clinic, specifically targeting NK$_1$R for the treatment of chemotherapy-induced nausea and vomiting (CINV)[25]. Over the past four years, two further molecules (i.e. netupitant and rolapitant) that do not share the common chemical scaffold of these earlier antagonists have been approved for use in the clinic for the same indication[17].

Extensive structure-activity relationship (SAR) studies performed over the course of three decades have revealed insights into overlapping and non-overlapping binding sites involved in recognition of peptide agonists and non-peptide antagonists in NK$_1$R[26–28]. However, until now, little has been known about the precise binding mode of small-molecule antagonists to NK$_1$R in the absence of a structure of this receptor. Since the prototypical antagonist CP-99,994 already displays many chemical features that are relatively preserved in further developed compounds we initially solved the structure of NK$_1$R in complex with this ligand. We then went on to co-crystallise NK$_1$R with two FDA-approved drugs—aprepitant and netupitant.

Here, we report three crystal structures of the human NK$_1$R bound to CP-99,994 and the clinically approved antagonists aprepitant and netupitant at 3.27, 2.40 and 2.20 Å resolution, respectively. These structures provide detailed and high-resolution structural insights into the molecular determinants of NK$_1$R antagonist recognition. The clinically approved antagonists are able to invoke structural rearrangements in the orthosteric binding pocket at the extracellular ends of helices V and VI and the extracellular loops (specifically ECL2) that govern the overall size and nature of the pocket, thereby acting to negatively modulate the receptor via an induced-fit binding mechanism. The observed high degree of plasticity in the NK$_1$R orthosteric binding pocket across the three structures reported here vastly improves our structural knowledge of NK$_1$R, explaining the different properties of current inhibitors and potentially facilitating the future development of ligands selectively targeting various NK receptors.

## Results

**Crystallisation of antagonist-bound NK$_1$R.** To improve protein expression, and ultimately the yield of NK$_1$R preparations, two consecutive rounds of directed evolution in *Saccharomyces cerevisiae* were initially performed on the human NK$_1$R[29]. One evolved receptor mutant (NK$_1$R-y04) was further thermo-stabilised in an antagonist-bound state through incorporation of four amino acid substitutions, L74$^{2.46}$A, A144$^{4.39}$L, A215$^{5.57}$L and K243$^{6.30}$A (Ballesteros and Weinstein numbering[30] denoted in superscript), leading to NK$_1$R$_S$ (Methods and Supplementary Figure 2a, b). To facilitate crystallisation in lipidic cubic phase, 11 residues (E227-H237) of the third intracellular loop (ICL3) were replaced by the thermostable PGS (*Pyrococcus abysii* glycogen synthase) domain[31]. The crystallised PGS fusion construct NK$_1$R$_{XTAL}$ is able to bind all co-crystallised antagonists with low nanomolar affinity. However, ligand affinity is reduced ∼10-fold when compared to the wild-type receptor, possibly due to an increased rigidity of the stabilised fusion construct (Supplementary Table 1). For crystallisation in lipidic cubic phase, NK$_1$R$_{XTAL}$ was purified from *Spodoptera frugiperda* (*Sf9*) insect cell membranes in the presence of either CP-99,994, aprepitant or netupitant. We then crystallised and determined three crystal structures of NK$_1$R bound to three different antagonists, complexed with CP-99,994 at 3.27 Å resolution, aprepitant at 2.40 Å resolution and finally with netupitant at 2.20 Å resolution (Table 1), with strong and unambiguous electron density for each antagonist present in the orthosteric site as well as key interaction residues of the receptor (Fig. 1d–f and Supplementary Figure 3d–i).

While NK$_1$R in complex with CP-99,994 crystallised in space group C222$_1$ (Supplementary Figure 4b, d–f), it is noteworthy that crystallisation of NK$_1$R with both aprepitant and netupitant consistently lead to better diffracting crystals in a different condition belonging to space group P2$_1$2$_1$2$_1$ (Supplementary Figure 4c, g–i). However, attempts to switch crystallisation conditions, i.e. crystallisation of NK$_1$R with CP-99,994 in those specific to netupitant or aprepitant, were not successful, indicating that the receptor conformations described here are specific to the ligands with which they are crystallised.

**Overall architecture of NK$_1$R.** Overall, NK$_1$R exhibits the canonical GPCR architecture comprising seven transmembrane helices (I–VII) with helix 8 lying parallel to the membrane plane (Fig. 1a–c). All intracellular and extracellular loops (ICLs and ECLs, respectively) are well resolved with the exception of ECL3. In NK$_1$R, similar to other structures of the β-branch of class A GPCRs, ECL2 forms an extended β-hairpin crossing above the orthosteric pocket and is anchored to the extracellular tip of helix III through a conserved disulfide bridge between C180$^{ECL2}$ and C105$^{3.25}$.

The structure of the transmembrane helical bundle of NK$_1$R is similar to those of other receptors from the β-branch of class A GPCRs bound to small-molecule antagonists, with root-mean-square deviations (RMSD) for backbone atoms of 1.3 Å to orexin 2 receptor[31] (OX$_2$R) (PDB ID 4S0V), 1.4 Å to neuropeptide Y Y1 receptor[32] (Y$_1$R) (PDB ID 5ZBH) and 2.5 Å to endothelin B receptor[33] (ET$_B$R) (PDB ID 5XPR) (Supplementary Figure 5a–i). In the three NK$_1$R structures, the highly conserved residue W261$^{6.48}$, which has been reported as the "toggle switch"

**Table 1 Data collection and refinement statistics**

|  | NK$_1$R:CP-99,994[a] (PDB 6HLL) | NK$_1$R:aprepitant[a] (PDB 6HLO) | NK$_1$R:netupitant[a] (PDB 6HLP) |
|---|---|---|---|
| *Data collection* |  |  |  |
| Space group | C222$_1$ | P2$_1$2$_1$2$_1$ | P2$_1$2$_1$2$_1$ |
| Cell dimensions |  |  |  |
| $a, b, c$ (Å) | 62.00, 122.73, 286.49 | 62.19, 76.45, 167.12 | 61.66, 76.57, 166.04 |
| $\alpha, \beta, \gamma$ (°) | 90.00, 90.00, 90.00 | 90.00, 90.00, 90.00 | 90.00, 90.00, 90.00 |
| Resolution (Å) | 47.88-3.27 (3.53-3.27)[b] | 48.24-2.40 (2.49-2.40)[b] | 49.50-2.20 (2.27-2.20)[b] |
| $R_{merge}$ | 0.689 (4.652) | 0.188 (2.879) | 0.161 (3.066) |
| $R_{pim}$ | 0.173 (1.225) | 0.052 (0.795) | 0.054 (1.035) |
| $I/\sigma(I)$ | 5.1 (1.1) | 12.8 (1.7) | 11.9 (1.3) |
| CC$_{1/2}$ | 0.990 (0.311) | 0.999 (0.753) | 0.998 (0.666) |
| Completeness (%) | 99.9 (99.6) | 100.0 (100.0) | 99.9 (99.4) |
| Redundancy | 32.2 (29.6) | 26.2 (26.4) | 18.5 (18.8) |
| *Refinement* |  |  |  |
| Resolution (Å) | 24.92-3.27 | 29.44-2.40 | 29.24-2.20 |
| No. of reflections (test set) | 17,325 (874) | 31,899 (1656) | 40,668 (1997) |
| $R_{work}/R_{free}$ | 0.225/0.275 | 0.201/0.229 | 0.204/0.227 |
| No. atoms |  |  |  |
| Protein | 3754 | 3832 | 3882 |
| Ligand | 22 | 37 | 42 |
| Water/ion/lipid | – | 355 | 475 |
| *B*-factors |  |  |  |
| Protein | 91.30 | 62.96 | 55.97 |
| Ligand | 87.89 | 54.13 | 51.20 |
| Water/ion/lipid | – | 85.90 | 82.10 |
| R.m.s. deviations |  |  |  |
| Bond lengths (Å) | 0.004 | 0.003 | 0.004 |
| Bond angles (°) | 0.911 | 0.578 | 0.699 |

[a]The structures of NK$_1$R in complex with CP-99,994, aprepitant and netupitant were solved using X-ray diffraction data from 6, 6, and 5 crystals, respectively
[b]Values in parentheses are for highest-resolution shell

important in triggering GPCR activation[34], is in a similar conformation to those observed in other inactive class A GPCR structures. Furthermore, the "ionic lock" involving R$^{3.50}$ of the D/ERY motif, making an intrahelical salt bridge with D$^{3.49}$, is present in all solved NK$_1$R structures. Taken together, the NK$_1$R structures in complex with different antagonists capture the inactive receptor conformation.

**The NK$_1$R orthosteric antagonist binding pocket**. In all three NK$_1$R co-structures, the antagonist small molecules are found buried within a largely hydrophobic cleft constituting the orthosteric binding pocket. The binding site is characterised by being elongated on a trajectory between helices II and IV and laterally constricted by side chains of helices III and VI (Fig. 1d–f and Supplementary Figure 6a–c). Although the overall shape of the NK$_1$R orthosteric binding pocket is itself unique, its depth and overall location within the transmembrane helical bundle is comparable to those in previously reported closely related structures, confirming that the non-peptide NK$_1$R small molecule antagonists occupy at least in part the typical drug-binding site of peptide-activated receptors (Supplementary Figure 5a–i).

Most high-affinity NK$_1$R-selective antagonists are characterised by a common pharmacophore. In general, this structural framework consists of a central six-membered ring with two (and in later-stage compounds three) bulky substituents (Fig. 2d–f). Henceforth, and for clarity, the central ring is termed the "core", the linker-attached bulky aromatic group (i.e. the methoxyphenyl group in CP-99,994 or di-trifluoromethylphenyl in all other antagonists) "arm 1" and the directly attached aromatic moiety (i.e. the phenyl in CP-99,994 or fluorophenyl in aprepitant or methylphenyl in netupitant) "arm 2". The additional variable cyclic substituent at the core present in aprepitant and netupitant is termed "arm 3". Since CP-99,994

represents the progenitor antagonist on which many of the later developed compounds were based (and as such makes only limited contact to the receptor) the interactions of CP-99,994 within the NK$_1$R binding site are described first.

CP-99,994 specifically interacts with only seven residues from helices III to VI and one residue from ECL2 in NK$_1$R (Fig. 2a and Supplementary Figure 3a). The core (2,3-*cis*-substituted-piperidine) is laterally wedged between the side chains of F268$^{6.55}$ and Q165$^{4.60}$ and capped by I182$^{ECL2}$. Q165$^{4.60}$ is positioned to concomitantly hydrogen-bond the piperidine core and the amine linker of arm 1. Mutation of Q165$^{4.60}$ to either alanine, glutamic acid or aspartic acid severely reduce the binding affinity of CP-99,994 (Fig. 2g). These data are in agreement with previous studies reporting the critical nature of this interaction in binding of CP-99,994 and other early non-peptide antagonists to NK$_1$R[35]. Mutation of F268$^{6.55}$ to alanine attenuates binding affinity more than 10-fold[36], highlighting the crucial nature of the hydrophobic stacking interactions with the small-molecule core moiety on this side of the orthosteric pocket.

Arm 1 of CP-99,994 extends deep into the receptor core, exploiting a large, almost exclusively lipophilic groove at the base of the orthosteric pocket formed by M291$^{7.39}$, M295$^{7.43}$, M81$^{2.53}$, W261$^{6.48}$, I204$^{5.46}$, F264$^{6.51}$ and P112$^{3.32}$. Within this region, the aromatic methoxyphenyl group of arm 1 is wedged between F264$^{6.51}$ and P112$^{3.32}$ and sits above a deep protrusion into the receptor core. The methoxy group points into a relatively spacious extension of this pocket comprised of residues from helices II, III and VII. Mutation of P112$^{3.32}$ to either aspartic acid or histidine results in a 4000-fold loss in binding affinity of CP-99,994 to NK$_1$R[36] with mutation of F264$^{6.51}$ to alanine displaying only a moderate (4-fold) impairment of binding[35], highlighting the importance of targeting this lipophilic local environment. However, the importance of this sub-pocket is underlined by the

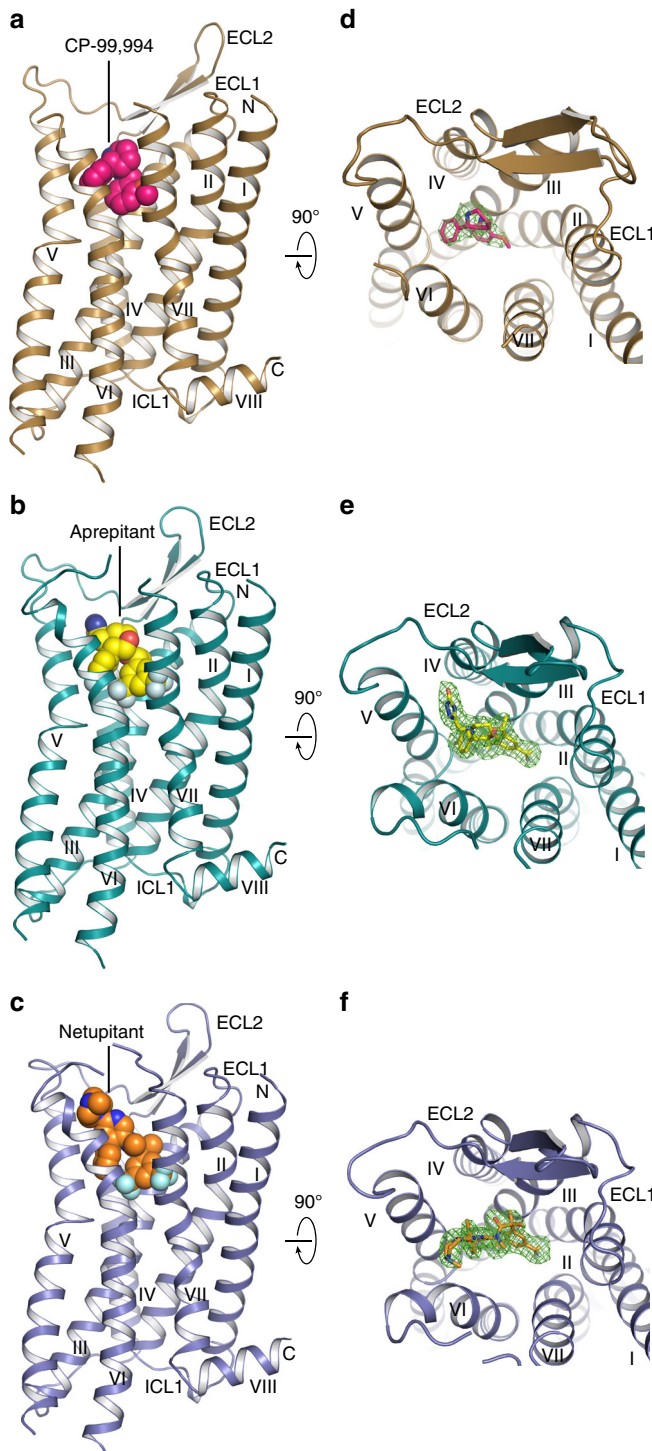

**Fig. 1** Overall structures of antagonist-bound NK$_1$R. **a–c** NK$_1$R in complex with CP-99,994 (**a**), aprepitant (**b**) and netupitant (**c**), viewed parallel to the membrane plane. The receptors are depicted by ribbons and coloured in brown, turquois and blue, respectively. The ligands CP-99,994, aprepitant and netupitant are shown as spheres and coloured in pink, yellow and orange, respectively. Oxygen, nitrogen and fluorine atoms of the ligands are highlighted in red, blue and grey, respectively. **d–f** NK$_1$R in complex with CP-99,994 (**d**), aprepitant (**e**) and netupitant (**f**), viewed from the extracellular space and coloured as in (**a–c**). The ligands are depicted as sticks. 2F$_o$-F$_c$ electron density maps of the ligands are shown in green mesh contoured at 1.0 σ

introduction of a more bulky tryptophan at this position leading to a >15-fold reduction in binding affinity for CP-99,994 (Fig. 2g).

Arm 2 of CP-99,994 targets a second lipophilic sub-pocket, making hydrophobic contacts to I113$^{3.33}$ and V200$^{5.42}$, and is capped on top by a face-on π-stack with H197$^{5.39}$. Mutation of H197$^{5.39}$ to alanine attenuates binding affinity 3.5-fold, yet preserving the aromatic character of this residue by virtue of mutation to phenylalanine results in only a <2-fold reduction in binding affinity to the small molecule, supporting this observed π-π interaction with the receptor.

Aprepitant, with its more extended structure due to its arm 3 substituent, engages in a host of additional interactions with NK$_1$R (Fig. 2b and Supplementary Figure 3b). The core of aprepitant (2,3,4-substituted-morpholine) is situated between F268$^{6.55}$ and Q165$^{4.60}$ and capped by a hydrophobic interaction with I182$^{ECL2}$ as in CP-99,994. However, Q165$^{4.60}$ does not contact the morpholine core, instead making hydrogen-bonds to the oxygen atom of the arm 1 ether linker and a nitrogen atom of the arm 3 substituent. The methyl substituent of the arm 1 ether linker then optimally targets a small lipophilic subpocket formed between N109$^{3.29}$, P112$^{3.32}$ and I113$^{3.33}$ on the surface of helix III.

The aromatic ring of arm 1 is shifted upwards and sideways towards helix II in comparison to CP-99,994. This arrangement allows it to optimally exploit the hydrophobic base of this part of the orthosteric pocket immediately above and adjacent to W261$^{6.48}$ by virtue of the 3,5-di-trifluoromethyl groups that straddle the aromatic side chain of this residue. It is possible that this direct engagement acts to prevent the activation-related motion of W261$^{6.48}$, thereby further stabilising NK$_1$R in the inactive conformation.

In a similar fashion to CP-99,994, arm 2 of aprepitant targets a lipophilic subpocket formed by residues H197$^{5.39}$, V200$^{5.42}$, T201$^{5.43}$, I204$^{5.46}$ and H265$^{6.52}$. However, the additional arm 3 triazolinone substituent of aprepitant is found creating an extended binding pocket (EBP) between the extracellular ends of helices IV, V and VI and ECL2. Aprepitant can thus engage an array of additional interactions with NK$_1$R. The arm 2 (fluorophenyl) and arm 3 (triazolinone) substituents of aprepitant participate in a π-stacking interaction with H197$^{5.39}$, while the carbonyl group of the triazolinone group reaches up to hydrogen-bond with the indole nitrogen of W184$^{ECL2}$, with E193$^{5.35}$ hydrogen-bonding to the N3 atom of the triazolinone ring. On the other side of the triazolinone ring, the carbonyl group of Q165$^{4.60}$ makes a hydrogen bond with N5 of this 5-membered ring. As a likely consequence of the additional network of interactions that aprepitant makes to the receptor, the affinity of this small molecule is much less affected by single point mutations in comparison to CP-99,994 (Fig. 2h). Mutation of Q165$^{4.60}$ to alanine, glutamic acid or aspartic acid only maximally impairs binding <5-fold, in agreement with earlier findings in which the mutant Q165$^{4.60}$A displayed a 10-fold decrease in binding affinity for the close aprepitant analogue L-742,694[23]. In contrast to CP-99,994, aprepitant therefore induces a conformational change of ECL2 and the extracellular ends of helix V and VI, highlighting an unexpected high degree of plasticity across this part of the receptor.

Both NK$_1$R antagonists described thus far are based on the chiral pharmacophore core of CP-99,994. In contrast, netupitant, which was disclosed in 2006 and clinically approved in 2014[37], belongs to a class of achiral, high-affinity NK$_1$R antagonists with an aromatic pyridine core[38]. In a similar fashion to aprepitant, netupitant adopts an elongated conformation inside the orthosteric binding site (Fig. 2c); however, it protrudes from the centre

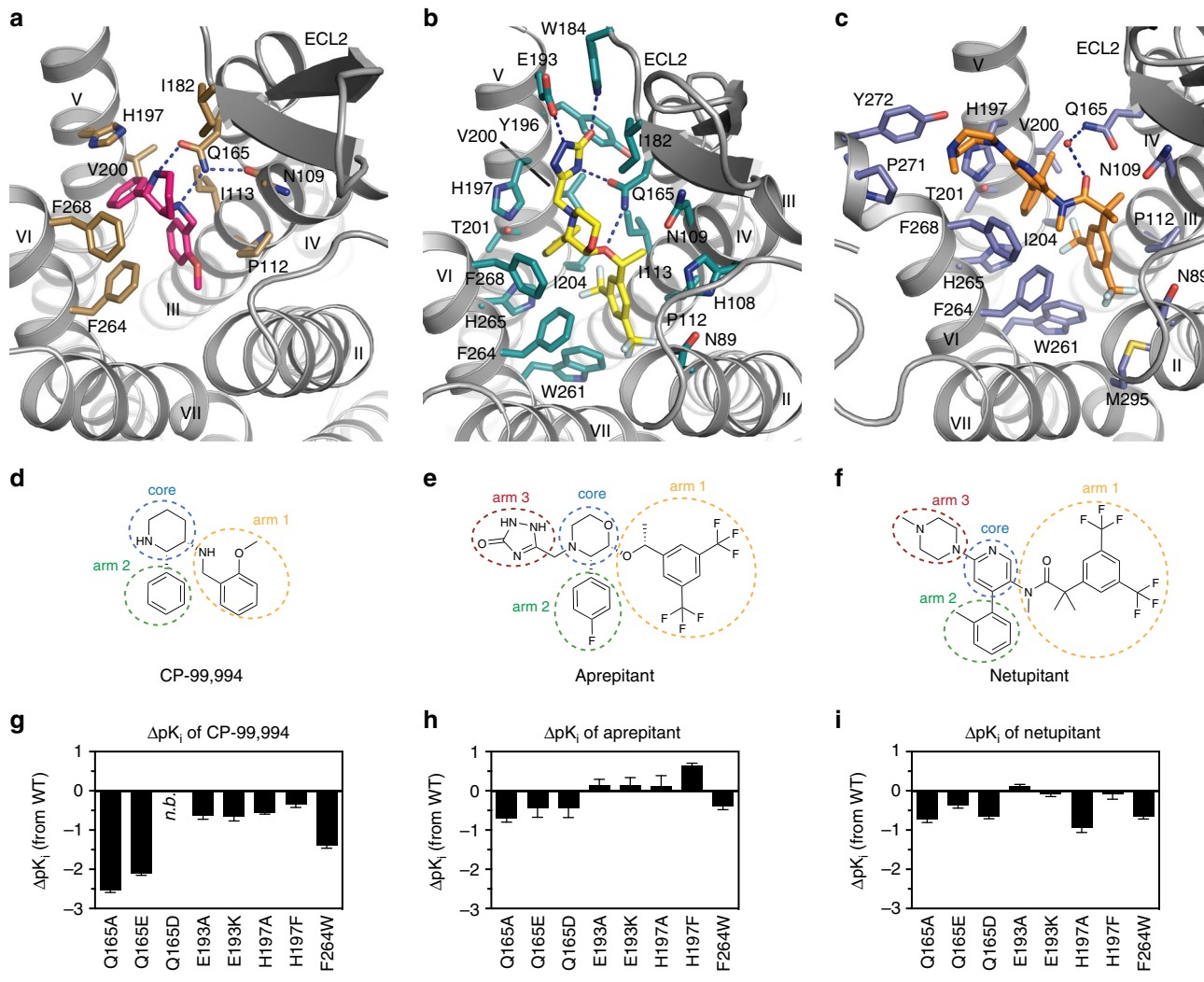

**Fig. 2** Drug-binding site of NK$_1$R. **a–c** Detailed interactions of CP-99,994 (**a**), aprepitant (**b**) and netupitant (**c**) with the receptor, viewed from the extracellular side from a position above helix I. The receptor backbone is shown in grey ribbon representation. Ligand and receptor residues within 4 Å of the respective antagonist are shown as sticks and are coloured as in Fig. 1. The ordered water involved in netupitant binding is depicted as a red sphere (**c**). Hydrogen bonds are indicated by dashed blue lines. **d–f** Chemical structures of CP-99,994 (**d**), aprepitant (**e**) and netupitant (**f**) with structural topology highlighted by coloured, dashed circles (core coloured in blue, arm 1 in yellow, arm 2 in green, arm 3 in red, respectively). **g–i** Antagonist affinity profiles of selected mutants in comparison to wild-type NK$_1$R. pK$_i$ values for each antagonist were derived from competition ligand-binding experiments (Supplementary Table 1). Bars represent differences in calculated affinity (pK$_i$) values for each mutant relative to the wild-type receptor for CP-99,994 (**g**), aprepitant (**h**), and netupitant (**i**). Data are shown as mean values ± s.e.m. from three to five independent experiments performed in duplicates. n.b. no binding. Source Data

of the orthosteric binding site towards extracellular space with both its core and arm 3 substituent. The di-trifluoromethylphenyl group of arm 1 and the methylphenyl group of arm 2 both target the same hydrophobic pockets as the equivalent substituents of the chiral antagonists. However, arm 2 is found to be in a more "upright" position. Major deviations in the positioning of the core region are observed within the orthosteric binding site, the linker region of arm 1, and the arm 3 substituent.

Due to the elongated linker of arm 1, the aromatic core of netupitant is pushed upwards and towards extracellular space from the centre of the pocket (towards helices V and VI) (Fig. 2c and Supplementary Figure 6c), while also moving sideways into closer proximity of F268[6.55]. This observed repositioning of the substituted pyridine towards helix V and in particular into closer proximity of H197[5.39] leads to an edge-to-face π-π interaction between this side chain and the aromatic core. Consequently, the importance of this aromatic interaction is reflected in a 10-fold loss in netupitant binding affinity upon mutation of H197[5.39] to

alanine. In contrast, preserving the aromatic nature at this position with a H197[5.39]F mutation leaves the binding affinity unchanged (Fig. 2i and Supplementary Table 1). In the netupitant-bound NK$_1$R structure, the side chain of Q165[4.60] is rotated by 180° (compared to all other structures), enabling the water-mediated coordination of the oxygen atom of the amide located in the arm 1 linker (Fig. 2c and Supplementary Figure 3c). Furthermore, due to the different ligand position within the binding site, arm 3 of netupitant targets a small hydrophobic groove formed between residues F268[6.55], P271[6.58] and Y272[6.59] at the extracellular tip of helix VI. The opening of this cleft between two helical turns is supported by a slight outward bending of the extracellular tip of helix VI towards the lipid bilayer. Overall, with the exception of the direct π-π interaction between the core and H197[5.39], the additional interactions between netupitant and NK$_1$R render the affinity of the antagonist less affected by single point mutations when compared to their influence on CP-99,994 binding affinity (Fig. 2i).

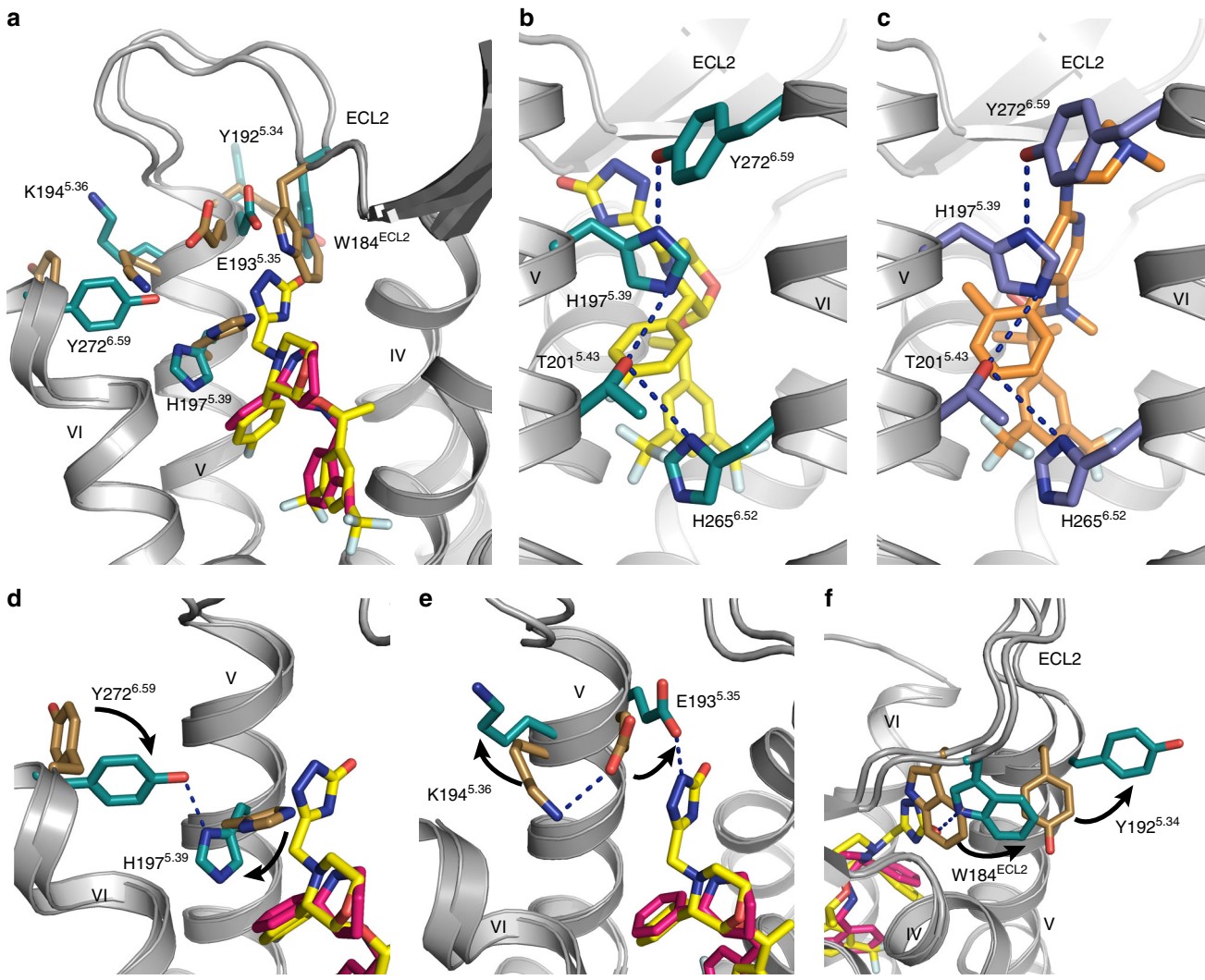

**Fig. 3** Conformational changes in NK$_1$R induced by clinically used antagonists. **a** Superimposition of the CP-99,994- and aprepitant-bound NK$_1$R structures, viewed from helix I. Residues of NK$_1$R with side-chain orientations differing between the two receptor structures as well as the antagonists are depicted as sticks, coloured as in Fig. 1. **b**, **c** Hydrogen bond network connecting the extracellular ends of helices V and VI in the aprepitant- (**b**) and netupitant-bound (**c**) NK$_1$R structures as viewed from the membrane plane. Hydrogen bonds are indicated as dashed blue lines. **d**–**f** Close-up views on residues with differing side-chain orientation in the CP-99,994- and the aprepitant-bound NK$_1$R structure. Side-chain rearrangements from the CP-99,994- to the aprepitant-bound conformation are indicated by black arrows

**Conformational changes induced by clinical NK$_1$R antagonists.**
The overall structure of netupitant-bound NK$_1$R is similar to that of the aprepitant-bound structure with an all-atom RMSD of ∼0.40 Å (Supplementary Figure 7a). However, both the aprepitant- and netupitant-bound NK$_1$R structures differ from the CP-99,994-bound structure with all-atom RMSDs of ∼0.70 Å. In comparison to the CP-99,994-bound receptor structure, aprepitant induces structural rearrangements in the extracellular regions of NK$_1$R, and thus modulates the extracellular molecular landscape of the receptor (Fig. 3a and Supplementary Figure 7b): The β-sheet of ECL2 is kinked away from the extracellular end of helix III by 4.1 Å (as measured between equivalent Cα atoms of T173$^{ECL2}$), thereby accommodating an inward movement of helix II and ECL1 by 1.7 Å (as measured between equivalent Cα atoms of H95$^{2.67}$) and 2.1 Å (as measured between equivalent Cα atoms of E97$^{ECL1}$), respectively. In addition, the β-hairpin loop is itself twisted towards the central axis of the receptor. This appears to be related to the observed *cis*-configuration of the peptide bond preceding P175$^{ECL2}$ located at the turn of the β-hairpin (Supplementary Figure 7c–e), while a *trans*-proline is found in

the CP-99,994-bound structure. Together, these differences lead to a slight contraction of the orthosteric binding pocket at the helix II–IV interface of the receptor when binding aprepitant or netupitant.

On the opposite side of the NK$_1$R orthosteric hemisphere, the extracellular end of helix V together with the C-terminal portion of ECL2 is shifted 2.4 Å away from the central axis of the receptor (as measured between equivalent Cα atoms of K190$^{ECL2}$). This shift is induced by the arm 3 substituent of aprepitant, which creates an EBP at the interface between helices IV, V and ECL2, thereby inducing significant side-chain rearrangements on this side of the receptor (Fig. 3a). The outward movement of the C-terminal part of ECL2 starts at W184$^{ECL2}$, which is pushed 3.5 Å out of the interhelical interface between helices IV and V by the triazolinone group of aprepitant (Fig. 3f) in comparison to the CP-99,994-bound structure. The outward push of W184$^{ECL2}$ is accommodated by a 90° rotation of Y192$^{5.34}$ towards the lipid bilayer.

Further rearrangements caused by aprepitant are observed at the extracellular portion of helices V and VI. Firstly, K194$^{5.36}$

is moved ~90° out of the orthosteric pocket, disengaging from E193[5.35] which is then free to hydrogen-bond the triazolinone ring of aprepitant (Fig. 3e). Secondly, the side chain of H197[5.39] is rotated by ~90° out of the orthosteric pocket, while Y272[6.59] at the top of helix VI is moved by ~90° into the interface between helices V and VI (Fig. 3d). This rearrangement of H197[5.39] and Y272[6.59] causes the formation of an extended interhelical hydrogen bond network connecting residues Y272[6.59], H197[5.39], T201[5.43] and H265[6.52], thereby cross-linking the extracellular ends of helices V and VI (Fig. 3b).

Although the arm 3 substituent of netupitant targets a part of the receptor that is distinct from aprepitant (see above), both antagonists are found to induce the same hydrogen bond network by virtue of the reorientation of H197[5.39] and Y272[6.59] (Fig. 3b, c). However, in contrast to aprepitant, in the case of netupitant, the reorientation of H197[5.39] is not caused solely by arm 3, but rather by the different binding pose of the compound within the orthosteric pocket, which shifts both the aromatic core and arm 2 into a position closer to helix V. This shift sterically precludes any other rotameric orientation of H197[5.39] except the one observed in the crystal structure found engaged with the extended hydrogen bond network.

Interestingly, H197[5.39] has previously been linked to play a role in insurmountable antagonism[23]. In that study, a single point mutation of H197[5.39] to serine was shown to alter the mode of antagonism of a close aprepitant analogue (L-742,694, Supplementary Figure 1) from insurmountable to surmountable while only moderately affecting binding affinity. Furthermore, it was demonstrated that tethering of the tips on helix V and VI via engineering of a high-affinity metal ion-binding site involving H197[5.39] and two histidine residues introduced at position E193[5.35] and Y272[6.59] rendered the receptor in an inactive conformation[39]. Finally, this region of helix V, and rearrangements of the hydrogen-bonding network between conserved serines on helix V (specifically residues 5.42, 5.43 and 5.46 which are a valine, threonine and isoleucine in NK₁R, respectively) by the catechol group of epinephrine in β1-AR and β2-AR further points to the importance of this region on the extracellular side of helix V in controlling the constellation of GPCR functional states[40,41].

It has long been acknowledged that interhelical hydrogen bonds have a strong influence on the conformational stability of membrane proteins[42–44]. The newly created hydrogen bond network, in the aprepitant-bound and netupitant-bound structures, tethers helix V to helix VI (Fig. 3b, c). In particular certain residues within the network have been shown to be critically involved in GPCR receptor activation[41,45,46]. We thus hypothesise that the reduction in conformational flexibility through engagement of the "histidine-lock" might represent a key driver for the observed insurmountable antagonism that some compounds such as aprepitant and netupitant elicit at NK₁R.

Once the "histidine-lock" is engaged by structural rearrangements induced by the mentioned insurmountable antagonists, the receptor would be present in the proposed slow reversible state[47,48], while in the absence of tethering helices V and VI with a hydrogen bond network only the fast reversible state would be populated (as observed in the CP-99,994-bound NK₁R structure). Moreover, this structural differentiation might also be the basis of the prolonged in vivo efficacy of compounds such as aprepitant and netupitant[49,50]: once the lock is engaged, the distinct conformational state of the receptor may facilitate rebinding of compounds without the need for conformational rearrangements, thereby potentially increasing the effective association rate.

**Structural basis of tight signaling control at NK₁R.** In the available high-resolution structures of inactive class A GPCRs where water molecules can be resolved, a conserved water-mediated hydrogen bond network is found to connect helices II, III, VI and VII[51,52] (Supplementary Figure 8b–e). This network is often clustered around a central sodium ion coordinated between the highly conserved D[2.50] and a polar residue on helix III (X[3.39]); concomitantly, sodium has been extensively described as a negative allosteric modulator stabilising the inactive receptor conformation of class A GPCRs[53,54].

D[2.50] is highly conserved (98%) across all class A GPCRs, with NK₁R being one of the very few exceptions where an aspartic acid is not present at this position[55]. In NK₁R, this position is occupied by a glutamate residue, E78[2.50] (Fig. 4a, b). Recent functional studies[56] suggest that this sequence variant is linked to the unusual lack of constitutive signaling in NK₁R[27,36,57]. The high resolution of the netupitant-bound NK₁R structure reported here now provides a structural rationale for this observation. E78[2.50] occupies a more central position in the water-mediated hydrogen bond network compared to D[2.50], making direct

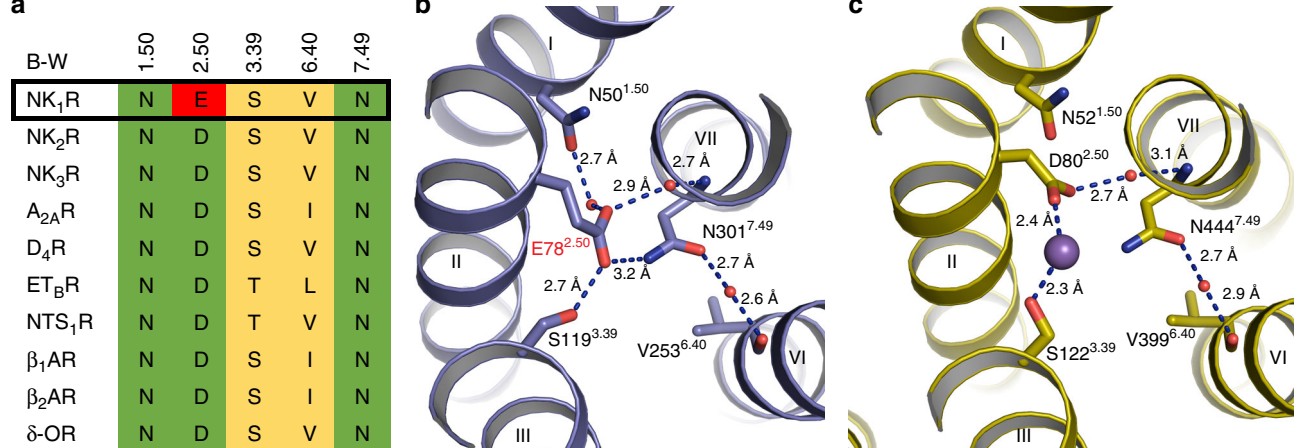

**Fig. 4** E78[2.50]-mediated interhelical hydrogen-bond network in the transmembrane core. **a** Amino acid sequence alignment of residues involved in the allosteric sodium ion-binding site in NK₁R and other class A GPCRs (Ballesteros-Weinstein numbers indicated). **b, c** Extracellular view on the hydrogen-bonding network in the transmembrane core of netupitant-bound NK₁R (**b**) and the dopamine D4 receptor (D₄R, PDB ID 5WIV) (**c**). The receptor backbone is shown in ribbon representations coloured blue and yellow, respectively. Ordered waters are shown as red spheres and hydrogen bonds are indicated by dashed blue lines. The sodium ion coordinated between D[2.50] and S[3.39] in the D₄R is shown as a purple sphere

hydrogen bonds to S119[3.39] and N301[7.49] of helices III and VII, respectively (Fig. 4b). This stands in contrast to previously reported GPCR structures, where the contact between the D[2.50] carboxyl group and the side chain of X[3.39] is mediated by the aforementioned sodium ion[58–61] (Supplementary Figure 8b–f) or potentially a water molecule[33]. Importantly, these features occur in addition to other conserved components of the network; for example, there is still a water-mediated hydrogen bond between the E78[2.50] side chain and the N301[7.49] backbone nitrogen, as well as another between N301[7.49] and V253[6.40], recapitulating the interactions observed in the high-resolution D[4]R antagonist structure[58] (Fig. 4c). Finally, a water-mediated hydrogen bond exists between helices I and II, through the side-chain oxygen atom of N50[1.50] and the carboxyl group of E78[2.50], similar to the interaction observed in several structures of other GPCRs such as the δ-OR[59], β[1]-AR[60] and A[2A]R[61] (Supplementary Figure 8c–e).

Therefore, it appears that the direct interactions mediated by the larger glutamate side chain of NK[1]R at position X[2.50] link the helical bundle of NK[1]R together more tightly than an aspartic acid would at the equivalent position, thus contributing to the stability of the inactive state of NK[1]R. Importantly, a recent study has shown that mutating E78[2.50] to aspartic acid results in a NK[1] receptor phenotype which is susceptible to a similar sodium concentration-dependent negative allosteric effect as many other class A GPCRs[56]. Thus, E78[2.50] appears to almost assume the functional role of the sodium ion in other GPCRs by virtue of one single extra carbon atom—and in fact, may further enhance, strengthen and expand the key sodium-mediated interactions in providing a structural basis for the observed lack of basal signaling in NK[1]R[56].

## Discussion

The crystal structures of NK[1]R with CP-99,994 (the progenitor compound) and with aprepitant and netupitant, in use in the clinic today for the treatment of CINV, reveal the structural basis of the successful high-throughput screening and medicinal chemistry campaigns directed towards this receptor spanning several decades. To recapitulate, modifications of arm 1 and arm 2 substituents lead to a more optimal interaction with the narrow orthosteric binding pocket, while the additional arm 3 substituent present in aprepitant and netupitant allows these compounds to induce a distinct receptor conformation by a potential induced-fit binding mechanism.

Additionally, the structures provide a possible molecular rationale for the observed insurmountable antagonism of aprepitant and netupitant. These compounds create an inter-helical hydrogen-bonding network at the extracellular ends of helices V and VI, engaging a "histidine lock" in NK[1]R across a region known a priori to be important in modulating activity for other class A GPCRs. Finally, a mechanistic basis for the lack of basal signaling of NK[1]R is provided by the network of hydrogen bonds mediated by the unique residue E78[2.50] in the helical core of the receptor, which overlaps the allosteric sodium binding site observed in many inactive class A GPCR structures to date.

In light of the research presented here, it is now possible to deploy structure-based drug-design methods for designing new small molecules directed to NK[1]R, for the aforementioned indications. Furthermore, while the residues in the lower half of the orthosteric pocket are largely conserved between NK[1]R, NK[2]R and NK[3]R, sequence homology weakens towards the top of helix V and VI between these closely related receptors. With careful molecular modelling, coupled to molecular dynamics simulations, the structure of NK[1]R now opens new avenues to design the next generation of selective antagonists for the closely related NK[2]R and NK[3]R, by specifically targeting this region at the top of

helices V and VI. The potential indications range from central neurological disorders including major depressive disorder and schizophrenia[62,63], to vasomotor symptoms associated with the menopause[64]. Additionally, endosome-targeted NK[1]R small-molecule antagonists have recently been reported as pain relief agents, presenting an alternative to opioid-based drugs[18]. Since endosomal targeting as of now requires the coupling of the antagonist to a lipophilic moiety, our results will aid the rational design of new drugs in this class. In summary, the reported NK[1]R structures greatly contribute to our knowledge and understanding of how this clinically relevant family of receptors may be antagonised, providing opportunities to improve in vivo efficacy and reduce phase II attrition rates in the clinic for small-molecules directed towards these receptors.

## Methods

**Generation of NK[1]R crystallisation construct.** From a pool of previously published NK[1]R mutants obtained by directed evolution in yeast for improved expression levels[29] using HiLyte Fluor 488-labeled Substance P, NK[1]R-y04 (V116[3.36]I, M181[ECL2]K, W224[5.66]R, ΔC336-407) was selected as a potential crystallisation candidate since it could be well purified from *Spodoptera frugiperda* (*Sf*9) insect cells as a monodisperse protein. However, NK[1]R-y04 displayed only limited thermal stability in the CPM assay[65] in complex with CP-99,994. To increase the thermostability of NK[1]R-y04 in an antagonist-bound state, selected amino acids within the transmembrane helical bundle were substituted with either alanine or leucine (if the amino acid was an alanine). The resulting single mutants were evaluated based on their gain in thermostability as evidenced by an increase in melting temperature (Tm) in the CPM assay when purified in the presence of CP-99,994. Single point-mutations which displayed the highest apparent gain in Tm (L74[2.46]A, A144[4.39]L, A215[5.57]L and K243[6.30]A) were subsequently combined yielding the thermostabilised quadruple mutant NK[1]R[S]. Furthermore, to aid crystallisation in lipidic cubic phase, eleven residues (E227-H237) of the ICL3 were replaced by the thermostable PGS (*Pyrococcus abysii* glycogen synthase) domain, yielding NK[1]R[XTAL]. The final construct was cloned into a modified pFL vector (MultiBac system, Geneva Biotech) resulting in an expression construct with a melittin signal sequence followed by a FLAG-tag, His[10]-tag and a human rhinovirus 3C protease cleavage site N-terminal to the receptor gene (all primers used in this study are listed in Supplementary Table 2).

**Expression and purification of NK[1]R.** Recombinant baculovirus was generated using the MultiBac expression system. The receptor expression cassettes were integrated into the DH10EMBacY baculovirus genome and the resulting bacmids were transfected into *Sf*9 in 6-well tissue culture plates (2 ml, density of $4 \times 10^5$ cells/ml) using 8 μl of Cellfectin II Reagent (Thermo Fisher Scientific) and Sf-900 II SFM medium (Thermo Fisher Scientific). Viral P0 stocks were harvested as the supernatant after 4 days and were subsequently amplified to obtain high-titer viral P1 stocks. For expression, *Sf*9 insect cells in Sf-900 II SFM medium were infected with P1 virus at a cell density of $3 \times 10^6$ cells/ml and a multiplicity of infection of 5. Expression was performed for 72 h at 27 °C under constant shaking. Cells were harvested by centrifugation, washed with PBS, frozen in liquid nitrogen and stored at −80 °C.

Insect cells expressing NK[1]R[XTAL] were lysed and receptor-containing membranes isolated by repeated Dounce homogenisation in hypotonic (10 mM HEPES pH 7.5, 20 mM KCl, 10 mM MgCl₂, 50 μg/ml Pefabloc SC (Carl Roth), 1 μg/ml Pepstatin A(Carl Roth)) and hypertonic buffer (10 mM HEPES pH 7.5, 20 mM KCl, 10 mM MgCl₂, 1.0 M NaCl, 50 μg/ml Pefabloc SC, 1 μg/ml Pepstatin A). Purified membranes were resuspended in 30 ml hypotonic buffer supplemented with 40 μM of the respective antagonist (CP-99,994 (Tocris)/aprepitant (Sigma Aldrich)/netupitant (Selleckchem)), frozen in liquid nitrogen and stored at −80 °C until further use.

Frozen membranes were thawed on ice, the respective ligand (CP-99,994, aprepitant or netupitant) was added to a final concentration of 80 μM and the suspension was incubated for 30 min while turning. 2 mg/ml iodoacetamide (Sigma Aldrich) was added to the solution followed by another 30 min of incubation. Subsequently, the receptor was solubilised in 30 mM HEPES pH 7.5, 500 mM NaCl, 10 mM KCl, 5 mM MgCl₂, 50 μg/ml Pefabloc SC, 1 μg/ml Pepstatin A, 1% (w/v) *n*-dodecyl-β-D-maltopyranoside (DDM, Anatrace) and 0.2% (w/v) cholesteryl hemisuccinate (CHS, Sigma Aldrich) at 4 °C for 3 h. Insoluble material was removed by ultra-centrifugation and the supernatant was incubated with TALON IMAC resin (GE Healthcare) at 4 °C overnight.

The receptor-bound resin was washed with 30 column volumes (CV) of Wash Buffer I (50 mM HEPES pH 7.5, 500 mM NaCl, 10 mM MgCl₂, 5 mM imidazole, 10% (v/v) glycerol, 1.0% (w/v) DDM, 0.2% (w/v) CHS, 8 mM ATP, 40 μM CP-99,994/20 μM aprepitant/20 μM netupitant) followed by 30 CV of Wash Buffer II (50 mM HEPES pH 7.5, 500 mM NaCl, 15 mM imidazole, 10% (v/v) glycerol, 0.05% (w/v) DDM, 0.01% (w/v) CHS, 40 μM CP-99,994/20 μM aprepitant/20 μM netupitant). Antagonist-bound NK[1]R was eluted step-wise with four column

volumes of Elution Buffer (50 mM HEPES pH 7.5, 500 mM NaCl, 250 mM imidazole, 10% (v/v) glycerol, 0.05% (w/v) DDM, 0.01% (w/v) CHS, 100 μM CP-99,994/50 μM aprepitant/50 μM netupitant). Protein-containing fractions were concentrated to 0.5 ml using a 100 kDa molecular weight cut-off Vivaspin 2 concentrator (Sartorius Stedim) and added to a PD MiniTrap G-25 column (GE Healthcare) equilibrated with G25 Buffer (50 mM HEPES pH 7.5, 500 mM NaCl, 10% (v/v) glycerol, 0.03% (w/v) DDM, 0.006% (w/v) CHS, 100 μM CP-99,994/50 μM aprepitant/50 μM netupitant) to remove imidazole. The complex was treated for 6 h with His-tagged 3C protease and PNGaseF (both prepared in-house) to remove the N-terminal affinity tags and deglycosylate the receptor. After incubation with Ni-NTA resin (GE Healthcare) overnight, cleaved receptor was collected as the flow-through and then concentrated to ~50–60 mg/ml with a 100 kDa molecular weight cut-off Vivaspin 2 concentrator. Protein concentrations were determined by absorbance at 280 nm on a Nanodrop 2000 spectrophotometer (Thermo Fisher Scientific). Protein purity and monodispersity were assessed by SDS-PAGE and analytical size-exclusion chromatography using a Sepax Nanofilm SEC-250 column.

**Crystallisation in lipidic cubic phase.** NK$_1$R was crystallised using the in meso method at 20 °C. For this purpose, concentrated protein (~50–60 mg/ml) was mixed with molten monoolein (Sigma Aldrich) supplemented with 10% (w/w) cholesterol (Sigma Aldrich) using the twin-syringe method. The final protein: lipid ratio was 40:60 (v/v). 40 nl boli were dispensed on 96-well glass bases with a 120 μm spacer (SWISSCI), overlaid with 800 nl precipitant solution using a Gryphon LCP crystallisation robot (Art Robbins Instruments) and sealed with a cover glass. In initial screens of CP-99,994-bound NK$_1$R, crystals appeared after less than 1 h in a broad range of conditions. Optimised crystals used for data collection were grown in a precipitant condition consisting of 100 mM MES pH 6.0, 31% (v/v) PEG400, 190–210 mM potassium acetate, 2.4% (v/v) 2,5-hexanediol and 50 μM CP-99,994. Aprepitant-bound NK$_1$R yielded much fewer initial crystallisation hits compared to CP-99,994-bound NK$_1$R. Optimised star-shaped crystals used for data collection of aprepitant-bound NK$_1$R were obtained in a condition consisting of 100 mM sodium citrate pH 6.0, 31% (v/v) PEG400, 50–70 mM MgCl$_2$ and 50 μM aprepitant. Crystals used for data collection of netupitant-bound NK$_1$R were obtained in a condition consisting of 100 mM sodium citrate pH 6.0, 31% (v/v) PEG400, 40–50 mM Mg(HCO$_2$)$_2$ and 50 μM netupitant. Single crystals were mounted with Dual-Thickness MicroMounts (MiTeGen) of appropriate size for data collection and cryo-cooled in liquid nitrogen without the addition of further cryoprotectant.

**Data collection and structure determination.** X-ray diffraction data were collected at the X06SA beamline at the Swiss Light Source (SLS) of the Paul Scherrer Institute (PSI, Villigen, Switzerland) using a beam size of 10 × 10 μm and an EIGER 16 M detector. Datasets for CP-99,994-bound NK$_1$R were collected using a beam attenuated to 10%, 0.1° of oscillation and 0.1 s exposure time. All other datasets were collected using a beam attenuated to 30%, 0.1° of oscillation and 0.05 s exposure time. Data from individual crystals were integrated using XDS[66]. Data merging and scaling was carried out using the program AIMLESS from the CCP4 suite[67,68]. Data collection statistics are reported in Table 1.

Initial phases were obtained by molecular replacement (MR) with the program Phaser[69] using the truncated OX2R transmembrane domain (PDB ID 4S0V) and the separated PGS fusion protein[31] as independent search models looking for one copy of each domain. Manual model building was performed in COOT[70] using sigma-A weighted 2m|F$_o$|-|DF$_c$|, m|F$_o$|-D|F$_c$| maps together with simulated-annealing and simple composite omit maps calculated using Phenix[71]. Initial refinement was carried out with REFMAC5[72] using maximum-likelihood restrained refinement in combination with the jelly-body protocol. Further and final stages of refinement were performed with Phenix.refine[73] with positional, individual isotropic B-factor refinement and TLS. The final refinement statistics are presented in Table 1. Co-ordinates and structure factors have been deposited in the worldwide Protein Data Bank under accession codes 6HLL, 6HLO and 6HLP for the CP-99,994-, aprepitant- and netupitant-bound NK$_1$R, respectively.

**Whole-cell ligand-binding assay.** HEK293T/17 cells (ATCC) were cultivated in Dulbecco's modified medium (Sigma) supplemented with 100 units/ml penicillin, 100 μg/ml streptomycin (Sigma) and 10% (v/v) foetal calf serum (BioConcept). Cells were maintained at 37 °C in a humidified atmosphere of 5% CO$_2$, 95% air. Transient transfections were performed with TransIT-293 (Mirus Bio) according to the manufacturer's instructions.

Ligand-binding experiments were performed on whole HEK293T cells for comparison of affinities for wild-type and receptor mutants using a homogeneous time-resolved fluorescence (HTRF) binding assay. Receptor mutants were generated by site-directed mutagenesis and cloned into a mammalian expression vector (pcDNA3.1(+)) containing an N-terminal SNAP-tag (Cisbio). HEK293T cells were transiently transfected with receptor constructs and seeded at 20,000 cells per well in poly-L-lysine-coated 384-well plates (Greiner). Forty-eight hours after transfection, cells were labelled with 50 nM SNAP-Lumi4-Tb (Cisbio) in assay buffer (20 mM HEPES pH 7.5, 100 mM NaCl, 3 mM MgCl$_2$ and 0.2% (w/v) nonfat milk) for 1.5 h at 37 °C. Cells were washed four times with

assay buffer and were then incubated for 2 h at RT in assay buffer containing fluorescently labelled peptide SP-HL488 (Substance P labelled with HiLyte Fluor 488 at Lys-3 (Anaspec))[29]. For competition binding, 20 nM of SP-HL488 tracer peptide and a concentration range of unlabelled antagonists as competitor were used. Fluorescence intensities were measured on an Infinite M1000 fluorescence plate reader (Tecan) with an excitation wavelength of 340 nm and emission wavelengths of 620 nm and 665 nm for Tb$^{3+}$ and the fluorophore HiLyte Fluor 488, respectively. The ratio of FRET-donor and acceptor fluorescence intensities (F665 nm/F620 nm) was calculated. Total binding was obtained in the absence of competitor, and nonspecific binding was determined in the presence of 100 μM unlabelled competitor. Data were normalised to the specific binding for each individual experiment and were analysed by global fitting to a one-site heterologous competition equation with the GraphPad Prism software (version 6.07, GraphPad). To obtain K$_i$ values, data were corrected for fluorescent ligand occupancy of each mutant with the Cheng-Prusoff equation as K$_i$ = IC$_{50}$/(1 + [fl. ligand]/K$_d$).

**Reporting summary**. Further information on experimental design is available in the Nature Research Reporting Summary linked to this article.

## Data availability
Atomic coordinates and structure factors have been deposited in the Protein Data Bank under accession codes 6HLL, 6HLO and 6HLP for the CP-99,994-, aprepitant- and netupitant-bound NK$_1$R, respectively. Data supporting the findings of this manuscript are available from the corresponding author upon reasonable request. The source data underlying Fig. 2 and Supplementary Table 1 are provided as a Source Data file.

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

## Acknowledgements

We thank B. Blattmann of the Protein Crystallisation Center at the University of Zurich, the staff of the X06SA beamline at the Paul Scherrer Institute for support during data collection and I. Berger at the European Molecular Biology Laboratory for providing us with baculovirus transfer vectors. We thank G. Meier and B. Aebli for support during protein production. We thank M. Hillenbrand for support during initial pharmacological characterisation. We would furthermore like to thank F. Zosel for critical reading of the manuscript. C.K. is the recipient of a fellowship of the German Academy of Sciences Leopoldina (*LPDS 2009-48*) and a Marie Curie fellowship of the European Commission (FP7-PEOPLE-2011-IEF #299208). This work was supported by Schweizerischer Nationalfonds Grants 31003A_153143, 31003A_182334 and KTI grant 18022.1 PFLS-LS, all to A.P.

## Author contributions

J.S. devised and carried out the mutagenesis and thermostabilisation of the receptor, designed and characterised crystallisation constructs, expressed, purified, crystallised the NK$_1$R-PGS fusion protein and harvested crystals. J.S. established the receptor purification and LCP crystallisation platforms. J.E. supported cloning, expression and crystal harvesting. J.S., J.E. and A.S.D. collected data. J.S., J.E., P.R. and A.S.D. processed the data, solved and refined the structures. C.K. performed ligand-binding experiments and analysed the data. M.S. performed directed evolution in yeast. Project management was carried out by J.S. and A.P. The manuscript was prepared by J.S., J.E., A.S.D. and A.P. All authors contributed to the final editing and approval of the manuscript.

## Additional information

**Competing interests:** P.R., M.S. and A.S.D. are employees of Sosei Heptares, a company with activities in the GPCR field. All other authors declare no competing interests.

