## [Peer Review File · Nature Communications]

Reviewers' comments:

Reviewer #1 (Remarks to the Author):

Schöppe and al., report the crystal structure of the human neurokinin 1 receptor (NK1) in complex with two clinically used antagonist, aprepitant and netupitant to high resolution. They also solved the structure of the progenitor antagonist CP-99,994. They described the ligand binding mode for the 3 molecules and elaborate on differences between CP-99,994, and the two very similar structure of NK1-bound aprepitant and NK1-bound netupitant. They report structural rearrangements in the extracellular regions of NK1 with the beta sheet of the ECL2 kinked away from the extracellular helix III by around 4Å compared to the CP-99,994-bound NK1 structure. Are there any contributions from crystal packing since the CP-99,994 and the two other ligands crystallised in two different space groups? Additionally, it is surprising to get the 2 different conformations for the proline in the beta-hairpin loop, cis and trans from the same receptor sample. Is the resolution good enough and the density excluding any ambiguity at the level of this proline P175ECL2?

The direct interaction between the E782.50 (TM2) and the S1193.39(TM3) represents a nice finding and provides a structural rationale for the reduced constitutive activity at NK1 receptor. This observation is compatible with the knowledge available about sodium binding site previously reported for several receptors. Is there any effect of Na⁺ on agonist or antagonist binding at NK1, as reported for other receptors?

E782.50 is just one turn above thermostabilising mutation L2.46A. L2.46 is also very well conserved across family A GPCR. Did the authors consider a possible contribution of this mutation on the E782.50 rotamer and the subsequent molecular contact observed with S1193.39? This research article, is very well written, and well presented and the high-resolution structures presented here will guide and facilitate the structure-based drug design for designing new small molecules directed to NK1R.

Reviewer #2 (Remarks to the Author):

This manuscript by Schoppe et al uses a thermostabilised NK1 receptor (containing 7 mutations and a fusion protein) to solve the structure of NK1 receptor bound to 2 clinically approved antagonists and a prototypical antagonist. The paper is well written and the structures are very well presented within the figures and supplementary data.

The main caveat with these structures and their interpretation is that the two clinical ligands have 10-16 fold reduction in affinity for the receptor construct used for crystallisation relative to the wildtype receptor (as shown in supplementary table 1), suggesting that the bound ligands within these structure may not bind exactly the same as in the unmodified receptor. Reduced affinity is acknowledged in the text stating "slightly reduced affinity", however I would suggest rewording this as a 10-fold loss in affinity is more than slightly reduced in my opinion.

Given the caveat listed above, it is important to verify the observations from the structure using additional experimental methods. The authors have addressed this nicely with mutagenesis within key binding site residues, which appear to support the contacts within the structures and some of which also differentiate between the binding of the three ligands. However, I would like to have seen some additional mutagenesis performed to confirm the binding pose of arm 3 in the aprepitant and netupitant bound structures, given this arm is not present in the CP compound and the contacts differ for the two clinical ligands. In addition, a key conclusion from the structures is that the clinically used antagonists induce a structural rearrangement in extracellular regions that lead to a contraction of helix II-IV interface. Given X-ray structures are static and the study uses a modified receptor that alters the binding affinities of the bound ligands, validating this finding using an alternative experimental method would strengthen the paper.

The authors have also suggested the packing of E2.50 may be responsible for the unusual lack of constitutive signalling for NK1R. To confirm if this is indeed the case, it would be useful to mutate this residue to an aspartic acid to verify that this induces constitutive signalling (as would be expected if the hypothesis put forward is correct) and to assess the ability of sodium to negatively regulate the wildtype receptor vs the E2.50D mutation.

The histidine lock mechanism, while supported partially by experimental data in the literature, is still speculative. Based on this, please change the wording of the last sentence in this section on page 12 to "may" not "will".

Figure 3 should be referenced when discussing the newly created hydrogen bond network stated on page 14.

In the methods section, can the authors state if a ligand was used for the directed evolution and if so which one. In addition, what is the degree of thermostability induced by the thermostabilising mutations and fusion protein?

For figures 1,2 and supplementary figure 2, electron density was shown for the ligands. It would also be useful to show the density for key side chains that interact with the ligands.

In supplementary figure 7, the panels are mislabelled relative to the legend.

To facilitate reading we have copied the reviewers' comments in italics and given our responses directly underneath in plain text.

Response to reviewers' comments:

We were of course very pleased by the favorable assessment of the reviewers, and we thank them for their insightful comments. Our response to the reviewers' comments and a detailed description of the changes we made in the manuscript are listed below. All changes are marked in green in the revised manuscript.

Reviewer #1:

Remarks to the Author:

Schöppe and al., report the crystal structure of the human neurokinin 1 receptor (NK1) in complex with two clinically used antagonist, aprepitant and netupitant to high resolution. They also solved the structure of the progenitor antagonist CP-99,994. They described the ligand binding mode for the 3 molecules and elaborate on differences between CP-99,994, and the two very similar structure of NK1-bound aprepitant and NK1-bound netupitant.

They report structural rearrangements in the extracellular regions of NK1 with the beta sheet of the ECL2 kinked away from the extracellular helix III by around 4Å compared to the CP-99,994-bound NK1 structure. Are there any contributions from crystal packing since the CP-99,994 and the two others ligands crystallised in two different space groups ?

Response: We draw the referee's attention to the following excerpt from the manuscript:

"While NK₁R in complex with CP-99,994 crystallised in space group C222₁ (Supplementary Fig. 4b,d-f), it is noteworthy that crystallisation of NK₁R with both aprepitant and netupitant consistently lead to better diffracting crystals in a different condition belonging to space group P2₁2₁2₁ (Supplementary Fig. 4c,g-i). However, attempts to switch crystallisation conditions, i.e. crystallisation of NK₁R with CP-99,994 in those specific to netupitant or aprepitant were not successful, indicating that the receptor conformations described here are specific to the ligands with which they are crystallised."

To expand: great efforts were made to crystallise NK₁R with CP-99,994 in the crystallisation conditions that gave rise to the higher resolution diffracting crystals obtained with aprepitant and netupitant, however, these efforts were unsuccessful. Additionally, it was not possible to crystallise NK₁R complexed with aprepitant or netupitant in the conditions that gave crystals complexed to CP-99,994. As these are protein preparations where the receptor is incubated with the ligand throughout

the process (and so bespoke preparations for each ligand) prior to crystallisation, it is reasonable to conclude that, whatever protein conformational changes occur in the receptor when complexed to each ligand, occur prior to entrance to the LCP matrix, crystallogensis and lattice formation. The fact that the conditions were / are not interchangeable between different ligands demonstrates precisely this, and so in short, protein conformational change happens prior to lattice formation.

As is shown in the supplemental figure which illustrates the packing of the two different crystal forms (Supplementary Figure 4d-i), ECL2 is implicated in a different set of packing interactions between the two space groups. For the C-centered orthorhombic system (CP-99,994 specific) the receptors pack head-to-head, and the extracellular sides mediate the lattice contacts along the C-axis; in the primitive orthorhombic system the extracellular side of the receptor forms lattice contacts with a PGS fusion of the neighbouring symmetry mate. Indeed, superposition of the two forms of NK₁R show that the alternate conformations cannot be sterically accommodated in the alternate lattice(s). It is therefore, extremely unlikely that the conformations of ECL2 observed are due to packing interactions / crystallisation artefacts, and in fact, these two different crystal forms provide valuable insight into how the conformation of ECL2 and the extracellular side of the receptor can be modulated by different ligands.

Additionally, it is surprising to get the 2 different conformations for the proline in the beta-hairpin loop, cis and trans from the same receptor sample. Is the resolution good enough and the density excluding any ambiguity at the level of this proline P175^{ECL2}?

Response: With regards to the *cis*-conformation of P175^{ECL2} reported in the netupitant and aprepitant structures, the high resolution obtained, parameter to observation ratio and most importantly the quality of the electron density across this region allows a *cis*-configuration to be modelled with certainty and is most preferable, as is shown in Supplementary Figure 7e. Modelling of the same configuration of this residue for the lower-resolution CP-99,994 structure returned density with negative peaks in the difference map around this region, and thus the *trans*-configuration was preferred, particularly in context of the geometry of the neighbouring residues. It should be remembered that this is not the same receptor sample, it is the same construct expressed and purified with different ligands, which, although thermostabilised, still possesses the orthosteric and extracellular flexibility to bind different chemical ligand series, and to undergo conformational changes therein.

The direct interaction between the E78^{2.50} (TM2) and the S119^{3.39} (TM3) represents a nice finding and provides a structural rationale for the reduced constitutive activity at NK1 receptor. This observation is compatible with the knowledge available about sodium binding site previously reported for several

receptors. Is there any effect of Na⁺ on agonist or antagonist binding at NK1, as reported for other receptors?

Response: We thank the reviewer for this comment. Interestingly, the influence of sodium on NK₁R has been previously investigated by Valentin-Hansen et al. (ref. 56), and it was found that the presence of sodium had no effect on Substance P binding to the wild-type NK₁R. However, upon mutation of E78^{2.50} to D, the presence of sodium impaired Substance P affinity ~10-fold, so sodium shows a similar negative allosteric effect in the E78^{2.50}D NK₁R mutant as observed for other class A GPCRs. We have referenced the mentioned publication at the beginning and end of this section and we have now added a sentence specifically stating this previous finding:

“Importantly, a recent study has shown that mutating E78^{2.50} to aspartic acid results in a NK₁ receptor phenotype which is susceptible to a similar sodium concentration-dependent negative allosteric effect as many other class A GPCRs.”

E78^{2.50} is just one turn above thermostabilising mutation L2.46A. L2.46 is also very well conserved across family A GPCR. Did the authors considered a possible contribution of this mutation on the E78^{2.50} rotamer and the subsequent molecular contact observed with S119^{3.39}?

Response: We have indeed considered this possibility and the reviewer is correct that we do not have information on the rotameric state of E78^{2.50} in the non-stabilised receptor, as this cannot be crystallographically observed. However, within the limitations of modelling, the positions of water molecules and hydrogen bonding distances in this network are identical to other high-resolution class A GPCR structures (as illustrated in the enclosed figure below). For participation in the hydrogen bonding network in NK₁R, E^{2.50} has to establish the observed rotamer, placing one sidechain oxygen in a position equal to that observed for D^{2.50} in all the structures of other receptors. We consider it thus more likely that the sidechain conformation of E^{2.50} in NK₁R is due to conserved interactions within the hydrogen bonding network, and thus present like this also in the wild-type receptor, rather than being induced by conformational freedom gained through the mutation L^{2.46}A. With one sidechain oxygen involved in the water-mediated hydrogen bonding network, the other sidechain oxygen of E^{2.50} is in hydrogen bonding distance of S^{3.39} only in the observed rotameric states of both residues. Importantly, however, in the crystal structure of P2Y₁₂ where a less bulky T^{2.46} is found, S^{3.39} adopts the identical rotamer conformation as in receptors comprising the conserved L^{2.46}. It is thus highly likely that the distinct rotamer conformation of S^{3.39} in NK₁R is induced by E^{2.50}.

In conclusion, another rotameric state of E^{2.50} would unlikely be able to participate in the observed hydrogen bonds. Furthermore, its evolutionary selection over the commonly found Asp-Na⁺ complex would not be plausible were it not for these extensive, sodium-independent interactions.

This research article, is very well written, and well presented and the high-resolution structures presented here will guide and facilitate the structure-based drug design for designing new small molecules directed to NK1R

Response: We thank the reviewer for this comment.

Reviewer #2:

This manuscript by Schoppe et al uses a thermostabilised NK1 receptor (containing 7 mutations and a fusion protein) to solve the structure of NK1 receptor bound to 2 clinically approved antagonists and a prototypical antagonist. The paper is well written and the structures are very well presented within the figures and supplementary data.

Response: We thank the reviewer for this comment.

The main caveat with these structures and their interpretation is that the two clinical ligands have 10-16 fold reduction in affinity for the receptor construct used for crystallisation relative to the wildtype receptor (as shown in supplementary table 1), suggesting that the bound ligands within these structure may not bind exactly the same as in the unmodified receptor. Reduced affinity is acknowledged in the text stating “slightly reduced affinity”, however I would suggest rewording this as a 10-fold loss in affinity is more than slightly reduced in my opinion.

Response: We have now more prominently stated the apparent loss in affinity in the main text: “However, ligand affinity is reduced ~10-fold when compared to the wild-type receptor, possibly due to an increased rigidity of the stabilized fusion construct (Supplementary Table 1).” For the raised concern regarding the precise binding mode of the antagonists please refer to our detailed answer below the next reviewer comment.

Given the caveat listed above, it is important to verify the observations from the structure using additional experimental methods. The authors have addressed this nicely with mutagenesis within key binding site residues, which appear to support the contacts within the structures and some of which also differentiate between the binding of the three ligands. However, I would like to have seen some additional mutagenesis performed to confirm the binding pose of arm 3 in the aprepitant and netupitant bound structures, given this arm is not present in the CP compound and the contacts differ for the two clinical ligands.

Response: We thank the reviewer for his appreciation of the performed mutagenesis to validate the binding site and his interest in the binding mode of the co-crystallized antagonists. We would like to stress that no thermostabilising mutations are located within the orthosteric binding pocket or in the extended binding pockets targeted by aprepitant or netupitant, respectively.

Non-peptide NK₁R antagonists have a particularly rich history in structure-activity-relationship studies performed by independent research groups in both academia and industry. Importantly, our

antagonist-bound NK₁R crystal structures are in very good agreement with these previously performed extensive experiments to characterize the binding site of small-molecule NK₁R antagonists.

To expand: both CP-99,994 and aprepitant originate from the first disclosed non-peptide antagonist CP-96,345. In the original publication disclosing CP-96,345 (Snider et al., 1991) (Ref. 19) it was already appreciated by the authors that the stereochemistry of the compound tested is immensely important for interaction with the receptor stating

“In contrast, the (2R,3R)-enantiomer had no significant effect on the SP-elicited relaxations at 5×10⁻⁷ M, the highest concentration at which the active enantiomer (CP-96,345) was tested.”

Thus, the precise stereochemistry of the core substituents of antagonists stemming from this lineage of development is of great importance. In a subsequent study (Fong et al., 1993, Nature 362:350-3), through mutagenesis of His197^{5.39} as well as analysis of several synthesised structural analogues of CP-96,345, the authors concluded that His197^{5.39} is involved in amino-aromatic interactions with the benzhydryl moiety of CP-96,345. The benzhydryl moiety of CP-96,345 is the equivalent chiral arm 2 position of later compounds such as the phenyl of CP-99,994 and the fluorophenyl of aprepitant which in our structures are in contact with His197^{5.39}.

The importance of Q165^{4.60} has been reported for CP-96,345 in a study similarly conducted as the one described above (Fong et al., 1994, J Biol Chem. 269:14957-61). Mutation of Q165 to either alanine, serine or asparagine impairs binding affinity and the authors again used structural analogues to probe the exact interaction position, concluding that one or more hydrogen bonds between Q165^{4.60} and the core and arm 1 linker region of CP-96,345 play a critical role in ligand binding. For CP-99,994 the importance of these hydrogen bond interactions has been separately reported through mutational studies (Greenfeder et al. 1998) (ref. 35). Our co-structure is fully consistent and now demonstrates how these hydrogen bonds mediated by Q165^{4.60} are absolutely essential to coordinate CP-99,994 in an optimal position within the orthosteric binding pocket.

Furthermore, the observed location of the arm 1 substituent of CP-99,994 within the narrow and deep base of the orthosteric pocket is supported by a study employing steric hinderance mutagenesis (Holst et al., 1998) (ref. 36) where the authors conclude:

“...steric hinderance mutagenesis strongly indicates that one population of nonpeptide antagonists bind in the deep pocket of the main ligand-binding crevice of the NK₁ receptor...”

Reconfirmed by our own mutagenesis data we are thus highly confident that the observed binding mode and orientation of CP-99,994 within the orthosteric pocket adequately reflects the binding of this compound to the wild-type receptor.

Consequently, the conserved pharmacophore features (core, arm 1 and arm 2) of aprepitant are very similarly positioned within the orthosteric pocket, an observation that is supported by an earlier report (Cascieri et al., 1997) (ref. 23) for a very close aprepitant analogue (L-742,694) stating:

“...these data indicate that L-742,694 binds to the same site as competitive tachykinin NK1 receptor antagonists and that addition of the triazolinone moiety does not alter the nature of its interaction with these three residues (Q165^{4.60}, His197^{5.39} and H265^{6.52})”.

We have furthermore assessed the binding position of the arm 3 substituent through the mutation of two amino acids which are in hydrogen-bond contact with the triazolinone. Additionally, as is now illustrated in the modified Supplementary Figure 3h, all interacting sidechains in this region are very well resolved.

Regarding the different arm 3 position of netupitant we would like to emphasize that this compound stems from an independent series of non-chiral small molecule NK₁R antagonists with an aromatic core region. The identical arm 1 substituent of netupitant (compared to aprepitant) binds similarly within the base of the orthosteric pocket, therefore this conserved molecular anchoring point largely dictates the positioning of the rest of the molecule. Importantly, the differing positioning of netupitant within the orthosteric pocket is further highlighted by netupitant's high sensitivity to mutation of H197^{5.39} to alanine, as described in the main text. Since the arm 3 of netupitant is attached to the aromatic core region by a single carbon-nitrogen bond, it thus has to be located in the region observed in our crystal structure.

In conclusion, our observed binding positions of the different small molecule antagonists bound to the engineered receptor construct are in very good agreement with a wealth of existing prior studies, encompassing both compound modifications and receptor modifications. As all these studies, as well as our own mutational analysis, were performed in a wild-type receptor background in the native phospholipid environment we are highly confident that the reported binding poses of all antagonists are biologically meaningful.

In addition, a key conclusion from the structures is that the clinically used antagonists induce a structural rearrangement in extracellular regions that lead to a contraction of helix II-IV interface. Given X-ray structures are static and the study uses a modified receptor that alters the binding affinities of the bound ligands, validating this finding using an alternative experimental method would strengthen the paper.

Response: We agree with the reviewer that it would be ideal to have an orthogonal method to measure the contraction of the helix II-IV interface. However, it would be extremely difficult to design spectroscopic probes that would unequivocally report on this. Realistically, this will only be possible to measure outside of crystallography once the NMR assignment of GPCRs has reached the necessary maturity. To further address this remark, we would like to refer the reviewer to our detailed answer to Reviewer #1 regarding crystallogenesis and the observed differences and similarities between the three structures.

The authors have also suggested the packing of E2.50 may be responsible for the unusual lack of constitutive signalling for NK1R. To confirm if this is indeed the case, it would be useful to mutate this residue to an aspartic acid to verify that this induces constitutive signalling (as would be expected if the hypothesis put forward is correct) and to assess the ability of sodium to negatively regulate the wildtype receptor vs the E2.50D mutation.

Response: As this remark was similarly raised by Reviewer #1, please refer to our answer to Reviewer #1 concerning the influence and previously performed mutagenesis studies of E2.50.

The histidine lock mechanism, while supported partially by experimental data in the literature, is still speculative. Based on this, please change the wording of the last sentence in this section on page 12 to “may” not “will”.

Response: As suggested by the reviewer we have changed the wording accordingly:
“...the distinct conformational state of the receptor may facilitate rebinding of compounds without the need for conformational rearrangements, thereby potentially increasing the effective association rate.”

Figure 3 should be referenced when discussing the newly created hydrogen bond network stated on page 14.

Response: We thank the reviewer for this comment and have now referenced the appropriate panels of Figure 3.

In the methods section, can the authors state if a ligand was used for the directed evolution and if so which one. In addition, what is the degree of thermostability induced by the thermostabilising mutations and fusion protein?

Response: We have added a more detailed description of the ligand used during the directed evolution in the methods section:

“From a pool of previously published NK₁R mutants obtained by directed evolution in yeast for improved expression levels²⁹ using HiLyte Fluor 488-labeled Substance P, NK₁R-y04 (V116^{3.361}, M181^{ECL2}K, W224^{5.66}R, ΔC336-407) was selected as a potential crystallisation candidate since it could be well purified from *Spodoptera frugiperda* (Sf9) insect cells as a monodisperse protein.”

The degree of thermostability induced by the four thermostabilising mutations as well as the fusion protein is shown in Supplementary Figure 2b.

For figures 1,2 and supplementary figure 2, electron density was shown for the ligands. It would also be useful to show the density for key side chains that interact with the ligands.

Response: We agree with the reviewer that this information would be useful and have therefore added panels g-i to Supplementary Figure 3 depicting the electron density for key interaction residues within the orthosteric pocket of each receptor-ligand complex.

In supplementary figure 7, the panels are mislabelled relative to the legend.

Response: We thank the reviewer for pointing this out and we have corrected the figure legend in the revised Supplement file.

REVIEWERS' COMMENTS:

Reviewer #1 (Remarks to the Author):

The authors have answered all comments and the manuscript is suitable for publication.

Reviewer #2 (Remarks to the Author):

The authors provide a very strong and compelling rebuttal and have addressed the majority of my concerns. With respect to the contraction of the helix II-IV interface, this could be explored potentially through molecular dynamics simulations, however I agree with the authors comments that demonstrating this via direct physical methods may be difficult to achieve with the current technology. While some additional work here would definitely strengthen the study, this study in its current form is worthy of publication.